# Density Functional Study to Investigate the Ability of (ZnS)*_n_* (*n* = 1–12) Clusters Removing Hg^0^, HgCl, and HgCl_2_ via Electron Localization Function and Non−Covalent Interactions Analyses

**DOI:** 10.3390/molecules28031214

**Published:** 2023-01-26

**Authors:** Zhimei Tian, Chongfu Song, Hai Wu

**Affiliations:** School of Chemistry and Materials Engineering, Fuyang Normal University, Fuyang 236037, China

**Keywords:** (ZnS)*_n_* clusters, Hg^0^, HgCl and HgCl_2_, electron localization function (ELF), non−covalent interactions (NCI) analysis, adsorption

## Abstract

In this study, the density functional theory is used to study the ability of (ZnS)*_n_* clusters to remove Hg^0^, HgCl, and HgCl_2_ and reveals that they can be absorbed on (ZnS)*_n_* clusters. According to electron localization function (ELF) and non−covalent interactions (NCI) analyses, the adsorption of Hg^0^ on (ZnS)*_n_* is physical adsorption and the adsorption ability of (ZnS)*_n_* for removing Hg^0^ is weak. When (ZnS)*_n_* adsorbs HgCl and HgCl_2_, two new Hg−S and Zn−Cl bonds form in the resultant clusters. An ELF analysis identifies the formation of Hg−S and Zn−Cl bonds in (ZnS)*_n_*HgCl and (ZnS)*_n_*HgCl_2_. A partial density of states and charge analysis confirm that as Hg^0^, HgCl, and HgCl_2_ approach (ZnS)*_n_* clusters, atomic orbitals in Hg and Zn, Hg and S, as well as Zn and Cl overlap and hybridize. Adsorption energies of HgCl and HgCl_2_ on (ZnS)*_n_* clusters are obviously bigger than those of Hg^0^, indicating that HgCl and HgCl_2_ adsorption on (ZnS)*_n_* clusters is much stronger than that of Hg^0^. By combining ELF analysis, NCI analysis, and adsorption energies, the adsorption of HgCl, and HgCl_2_ on (ZnS)*_n_* clusters can be classified as chemical adsorption. The adsorption ability of (ZnS)*_n_* clusters for removing HgCl and HgCl_2_ is higher than that of Hg^0^.

## 1. Introduction

Mercury can cause serious damage to the human body and ecosystems, and it is one of the most harmful pollutants in the environment [1,2]. Mercury in the atmosphere is mainly derived from flue gases released from power plants. There are three main forms of mercury in flue gas, which are divalent mercury (Hg^2+^), particulate bound mercury (Hg_P_), and elemental mercury (Hg^0^) [2,3]. Hg^2+^ and Hg_P_ can be captured and removed by existing pollution control equipment. However, it is difficult to remove Hg^0^ with existing control devices due to its low melting point, low solubility, and high volatility [4,5,6]. Therefore, how to effectively remove Hg^0^ has become a major challenge to control flue gas mercury emissions. Recently, in industry, several technologies have been developed to prevent mercury emission from coal combustion, among which the most widely commercialized method is activated carbon injection before the electrostatic precipitator [7,8,9]. When the temperature of flue gas decreases, mercury is oxidized to HgCl_2_ due to the large amount of chlorine in the pulverized coal. Meiji found that Hg^0^ and HgCl_2_ coexisted in flue gas [10]. Mercury chlorination is generally considered to be the main mercury conversion mechanism in coal−fired flue gas. According to a study by Carpi, hydrogen chloride and other contaminants can influence the distribution condition of atomic and divalent mercury [11]. The higher the concentration of chlorine, the more easily atomic mercury is oxidized to mercury chloride [12].

Due to the limited adsorption performance of activated carbon, its operating cost increases and the efficiency fluctuates significantly when temperature changes. Thus, there is an urgent need to develop non−carbon adsorbents to effectively and economically reduce Hg^0^, HgCl, and HgCl_2_ pollution in flue gas. In recent experimental studies, environmentally benign zinc sulfide that consisted entirely of “active” sites has demonstrated the effectiveness of mercury capture from flue gas. Recent studies have shown that zinc sulfide had good mercury trapping performance [8,13,14]. As compared with traditional activated carbon, ZnS with nanostructure has shown good adsorption performance in terms of Hg^0^ adsorption capacity and absorption rate [8,15]. Recent studies have shown that the mercury capture performance of zinc sulfide was due to the interaction between S^2−^ and Hg^0^ [16]. Several theoretical studies have focused on the interaction between ZnS surface and Hg^0^ [8,16]. The binding mechanism of mercury species and typical flue gas components on the surface of ZnS(110) have been studied by using density functional theory. It was revealed that Hg^0^ could be chemically adsorbed on the surface of ZnS [8,13]. The active Zn sites were beneficial to excite the electrons on the outer shell of Hg^0^. Yang et al. explored the influence of different crystal forms of ZnS nanomaterials for the adsorption of elemental mercury [16], and they revealed that the removal performance of Hg^0^ depended on the specific surface area and S active site.

Clusters are good nanomaterials to absorb Hg^0^. It has been reported that neutral and charged noble metal clusters, such as Pd*_n_* [17,18], Au*_n_* [19], and Ag*_n_* clusters [20], could absorb Hg^0^. Recently, the absorption of Hg^0^ on (CuS)*_n_* (*n* = 1–12) clusters has been reported [21] and the absorption mechanism of Hg^0^ on (CuS)*_n_* clusters was revealed. There are many studies on the structures and properties of (ZnS)*_n_* clusters [22,23]. As far as we know, all studies on the adsorption of Hg^0^, HgCl, and HgCl_2_ by ZnS have focused on the surface of ZnS(110) nanomaterials. However, previous studies have paid less attention to the mechanism of removing Hg^0^, HgCl, and HgCl_2_ from ZnS clusters, which is also important for the modification of catalysts. Different mercury species including Hg^0^, HgCl, and HgCl_2_ are used in our work to understand the adsorption ability of (ZnS)*_n_* clusters. According to reports in the literature, the ZnS(110) surface has good removal ability of Hg^0^, HgCl, and HgCl_2_ [8,24]. The question is whether (ZnS)*_n_* clusters have the ability to remove Hg^0^, HgCl, and HgCl_2_? If (ZnS)*_n_* clusters can absorb Hg^0^, HgCl, and HgCl_2_, what are the bond properties between (ZnS)*_n_* clusters and Hg^0^, HgCl, and HgCl_2_? What is the difference between (ZnS)*_n_* clusters adsorption and ZnS(110) surface adsorption? With these questions in mind, here, the ability of (ZnS)*_n_* (*n* = 1–12) clusters removing Hg^0^, HgCl, and HgCl_2_ has been studied by density functional theory. First, the structures of (ZnS)*_n_* clusters are obtained. Then, the structures of the adsorbed products (ZnS)*_n_*Hg^0^, (ZnS)*_n_*HgCl, and (ZnS)*_n_*HgCl_2_ are obtained. Electron localization function (ELF) analysis and non−covalent interactions (NCI) analysis methods are used to study the bond properties between (ZnS)*_n_* clusters and Hg^0^, HgCl, and HgCl_2_. The adsorption energies are calculated. The partial density of states of parts of supported clusters are studied to obtain the orbital information.

## 2. Results and Discussion

### 2.1. Geometries of (ZnS)_n_ (n = 1–12) Clusters

The global minimum (GM) and low−lying structures of (ZnS)*_n_* (*n* = 1–12) clusters are displayed in Figure 1. 1a is a linear structure in *C_∞v_* with the Zn−S bond length being 2.05 Å. Here, the bond length of Zn−S is in the range of reported Zn−S bond lengths (2.05–2.13 Å) [22,25,26]. 2a is a rhombus *D*_2h_ structure. 2a−1 lies 1.27 eV higher than 2a, in which one S−S bond and one Zn−Zn bond are present. 3a is a triangle structure in *D*_3h_ symmetry and 3a−1 is a three−dimensional structure. 4a is composed of four S−Zn−S units, which is a rectangle structure in *D*_4h_ symmetry. 4a−1 and 4a−2 are both much higher in energy than 4a. Small size clusters (*n* = 1–4) have planar geometries, which is in accordance with previous studies [26,27,28]. (ZnS)*_n_* (*n* = 1–4) clusters adopt planar ring structures, in which Zn and S atoms alternate with the coordination number of each Zn and S being 2. 5a is a *C_s_* symmetry structure with five S−Zn−S units. 5a−1 lies 0.15 eV higher. 5a−2 lies much higher and its relative energy is 1.35 eV. 6a is in a hollow cage conformation composed of two triangular Zn_3_S_3_ structures. 6a−1 has been previously predicted to be the GM structure for (ZnS)_6_ [25]. Here, 6a−1 is identical to that reported in the literature using the density functional formalism and projector augmented wave method, which is a cage−like structure composed of four Zn_3_S_3_ and two Zn_2_S_2_ units. Our predicted GM (6a) for (ZnS)_6_ is 0.63 eV lower in energy than that in the literature [25]. 6a−2 is a crown structure, and its relative energy is 1.82 eV with respect to 6a. In 1a−6a, each S atom is coordinated with two neighboring Zn atoms. 7a is a cage structure in *C*_3_*_v_* symmetry, which includes four Zn_4_S_4_ and two Zn−S edge−sharing Zn_6_S_6_ units. 7a−1 and 7a−2 both lie much higher than 7a. 8a is a cage structure, and it is composed of four Zn_3_S_3_ and six Zn_2_S_2_ units. 8a−1 consists of two Zn_4_S_4_ rings, and its energy is 1.05 eV. The energy of 8a−2 is 1.13 eV. 9a is composed of two quasi−planar Zn_3_S_3_, three triangular six−number Zn_3_S, and six Zn_2_S_2_ units. 10a is a cage structure, which is composed of six Zn_3_S_3_ and three Zn_2_S_2_ units. The energy of 10a−1 is 0.08 eV higher, and it is composed of two pairs of Zn−S edge sharing Zn_3_S_3_ cells, and each cell is composed of two Zn_3_S_3_.

Two Zn_5_S_5_ ring cells constitute 10a−2 and its energy is 2.48 eV. 11a and 12a are both cage structures. 11a is composed of seven triangle Zn_3_S_3_ and four Zn_3_S_3_ units. The energy of 11a−1 is 0.68 eV. 12a in *T*_h_ symmetry has eight Zn_3_S_3_ and four Zn_2_S_2_ units, which is identical to the structure in the literature [29]. 12a−1 lies much higher than 12a. 10a, 11a, and 12a are the so−called “bubble clusters”, and Zn and S atoms in them connect with three neighboring atoms [30]. Thus, the coordination number of S atoms increase from two to three as the cluster changes from ring structure to hollow cage structure.

The global minimum structures (GMs) at the PBE0−D3BJ/def2−TZVP level of theory for (ZnS)*_n_*Hg, (ZnS)*_n_*HgCl, and (ZnS)*_n_*HgCl_2_ are shown in Figure 2. From the figure, the structure of 1b is an almost a linear structure, and the Hg atom is bound to the Zn atom. The formed Zn−Hg bond in 1b is 2.63 Å. In 2b, the Zn−Hg bond is 2.68 Å and Hg connects to the Zn atom. In 3b, three Zn atoms are located in the middle of three S−Zn−S units. The Hg atom connects to one of the three Zn atoms in the (ZnS)_3_ cluster and the Zn−Hg bond is 2.84 Å. The Zn−Hg bond in 4b is 3.01 Å. It is worth mentioning that Hg adsorbed on (ZnS)_4_ cluster has been studied recently and it was found that the Hg atom also bound to the Zn atom [21]. However, the global minimum structure of the (ZnS)_4_ cluster in the literature lies 0.27 eV higher relative to the GM structure reported here. In 5b, the Hg atom is located above the (ZnS)_5_ cluster, forming three Zn−Hg bonds, and the lengths of them are 3.41, 3.41 and 3.42 Å, respectively. The Hg atom in 6b is on the top of one Zn_3_S_3_ unit, forming three Zn−Hg bonds, which are all 4.03 Å. The Hg^0^ position in 6b is similar to Hg^0^ adsorbed on the CuS(001)−Cu/S surface [31]. In 5b and 6b, there are also Hg−S bonds present beside Zn−Hg bonds. In other words, Hg^0^ is stabilized by multi−interactions with neighboring Zn and S atoms. In 7b, the Hg atom connects to the (ZnS)_7_ cluster by the Hg atom and the Zn atom in one of (ZnS)_3_ units, and the forming Zn−Hg bond is 2.75 Å. The Hg atom in 8b links to the Zn atom at the edge of the (ZnS)_8_ cluster and the forming Zn−Hg bond is 2.80 Å. The forming Zn−Hg bonds in 9b, 10b, 11b, and 12b are 2.80, 2.85, 2.88, and 2.94 Å, respectively. From the above analysis, it is clear that the Hg atom interacts with the Zn atom in the (ZnS)*_n_* cluster forming a Zn−Hg bond. The Zn−Hg bond lengths in 1b−3b and 7b−12b are comparable to that in the Hg^0^ binding on the ZnS(110) system (2.97 Å) [8].

In 1c, Hg connects to S, while Cl bonds to Zn, and Hg−S is 2.50 Å and Zn−Cl is 2.10 Å. In 2c, the Cl atom links to the Zn atom through the Zn−Cl bond, whereas Hg interacts with S through the Hg−S bond. In 2c, the rectangle Zn_2_S_2_ conformation is damaged when HgCl goes near the (ZnS)_2_ cluster. It should be noted that Hg and Cl are parted with each other in 1c and 2c. When the HgCl molecule moves to the (ZnS)_3_ cluster, one of the Zn−S bonds break to form 3c. In 3c, the Zn−Cl and Hg−S bonds are 2.24 and 2.58 Å, respectively. Hg and Cl interact with S and Zn in 4a, respectively, to form 4c. As compared with **5a**, the structure of (ZnS)_5_ unit in **5c** changes. One S−Zn−S angle in 5c is as small as 124.3°. As compared with 6a, (ZnS)_6_ in 6c is similar to 6a. 6c forms from the Hg and Cl atoms interacting with the S and Zn atoms in different (ZnS)_3_ units of 6a. In 6c, Hg−S is 2.60 Å and Zn−Cl is 2.24 Å. When adding the HgCl molecule to 7a, the resultant molecule is 7c. The (ZnS)_7_ unit in 7c is similar to 7a. The Hg and Cl atoms interact with the S and Cl atoms in 7c, respectively. The Hg and Cl atoms form two six−number structures in 7c, and both of the six−number structures are Zn_2_S_2_HgCl. In 8c, the HgCl molecule and the neighboring Zn and S atoms form one eight−number unit and one six−number unit. When the HgCl molecule approaches 9a, one of the Zn−S bonds on the edge breaks to form Hg−S and Zn−Cl bonds. Two six−number Hg−S−Zn−S−Zn−Cl units form in 9c. In 10c and 11c, the Hg and Cl atoms of HgCl form two six−number ring structures, which are Hg−Cl−Zn−S−Zn−S units. When adding the HgCl molecule to 12a, two eight−number ring structures form in 12c. From the above analysis, it can be seen that when HgCl is adsorbed on (ZnS)*_n_* clusters, Hg in HgCl interacts with one S atom, and Cl in HgCl interacts with one Zn atom in the (ZnS)*_n_* cluster. The adsorption of HgCl on (ZnS)*_n_* clusters is different from that of HgCl on the ZnS(110) surface [8]. When adsorbing HgCl on the ZnS(100) surface, the Hg and Cl atoms of HgCl bind to two Zn atoms of the ZnS(100) surface [8].

In 1d, the Zn−Cl, Hg−S, and Hg−Cl bond lengths are 2.09, 2.30, and 2.28 Å, respectively. 2d is a strange structure. When the HgCl_2_ molecule approaches 1a, the HgCl_2_ molecule disintegrates to Hg, Cl, and Cl atoms. One Zn−Hg bond is present in 2d. The S−Cl bond length is 2.05 Å. 3d is similar to 3c. 3d can be regarded as one Cl atom adding to 3c. One Hg−Cl bond elongates to 3.02 Å, much bigger than the other Hg−Cl bond (2.27 Å). In 4d, one Cl atom is located above the Zn_4_S_4_ unit forming two Zn−Cl bonds. The Hg atom in the HgCl_2_ molecule interacts with the S atom through the Hg−S bond, which is 2.32 Å. The two Hg−Cl bonds in 4d are 3.66 and 2.27 Å, respectively. 5d is a cage structure, in which Hg, Cl, and Cl atoms in HgCl_2_ participate in forming a ring structure with neighboring Zn and S atoms. When the HgCl_2_ molecule approaches 6a, the Cl atom and one Zn atom in the Zn_3_S_3_ unit form a Zn−Cl bond, while the Hg atom and one S atom in another Zn_3_S_3_ unit forms an Hg−S bond. The Zn−Cl and Hg−S bonds are 2.21 and 2.36 Å, respectively. In 7d, Hg−S is 2.35 Å and Zn−Cl is 2.18 Å. In 8d, one Zn−S bond cracks to form Zn−Cl and Hg−S bonds when the HgCl_2_ molecule appears. Similarly, one Zn−S bond breaks to form Zn−Cl and Hg−S bonds, which are 2.17 and 2.34 Å, respectively. In 9d, one Zn−S bond breaks to form Zn−Cl and Hg−S bonds with the HgCl_2_ molecule. In 9d, the newly formed Zn−Cl bond is 2.20 Å and Hg−S is 2.35 Å. In 10d, the Zn−Cl and Hg−S bonds are also 2.20 and 2.35 Å, respectively. The main conformation of the (ZnS)_11_ unit in 11d does not change much as compared with 11a. The new formed bond lengths of Hg−S and Zn−Cl are identical in 10d and 11d. In 12d, the broken Zn−S bond is elongated to 3.98 Å, and the newly formed Zn−Cl and Hg−S bonds are 2.22 and 2.36 Å, respectively. The binding conformation of HgCl_2_ adsorbed by (ZnS)*_n_* clusters is similar to HgCl_2_ on the Fe_3_O_4_(111) surface [32]. When the HgCl_2_ molecule is adsorbed on the (ZnS)*_n_* cluster, HgCl_2_ decomposes into HgCl and Cl, then Hg in HgCl parts, and the separate Cl atoms bind with S and Zn atoms, respectively. However, when HgCl_2_ is adsorbed on the ZnS(110) surface, the Hg atom binds with the S atoms, and two Cl atoms bind with two Zn atoms. Here, when HgCl_2_ is adsorbed on the (ZnS)*_n_* cluster, only one Cl atom is bound to the Zn atom.

Based on the above analysis, when Hg^0^ approaches the (ZnS)*_n_* cluster, one Zn−Hg bond forms and the main frame of the (ZnS)*_n_* cluster remains. Hg^0^ prefers the Zn−Hg bond to the Hg−S bond. When the (ZnS)*_n_* cluster adsorb the HgCl molecule, one Zn−S bond in (ZnS)*_n_* cracks, and then Hg−S and Zn−Cl bonds form. When the HgCl_2_ molecule is absorbed by the (ZnS)*_n_* cluster, one Zn−S bond breaks, meanwhile Hg−S and Zn−Cl bonds form.

### 2.2. Adsorption Energy

The interactions among Hg^0^, HgCl, HgCl_2_, and (ZnS)*_n_* clusters are investigated to obtain the removal ability of (ZnS)*_n_* clusters. If *E*_ad_ is negative, the adsorption reaction is exothermic. The more negative *E*_ad_ is, the stronger the interaction between adsorbate and substrate. Generally, if *E*_ad_ is less than −29.8 kJ/mol, it is physical adsorption; if *E*_ad_ is higher than −50.0 kJ/mol, it is chemical adsorption [33]. The *E*_ad_ values of Hg^0^, HgCl, and HgCl_2_ with (ZnS)*_n_* (*n* = 1–12) clusters are calculated. *E*_ad_ values versus cluster sizes *n* are given in Figure 3.

From Figure 3a, the *E*_ad_ values of Hg^0^ on (ZnS)*_n_* (*n* = 1–12) clusters range from −13.34 to −4.04 kcal/mol. From the adsorption energy point of view, the adsorption of Hg^0^ on (ZnS)*_n_* clusters is physical and weak chemical adsorption. From Figure 3a, the *E*_ad_ of Hg^0^ decreases with an increase in *n*. At *n* = 1 and 4, the adsorption energies are the maximum and minimum of all sizes, which are −13.33 and −4.04 kcal/mol, respectively. At *n* = 5 and 7, *n* is larger than that of the adjacent sizes, which are −9.62 and −9.04 kcal/mol, respectively. A previous study has revealed that *n* for Hg^0^ on the ZnS(110) surface is −21.00 kcal/mol [8]. Thus, it is obvious that the adsorption of Hg^0^ over (ZnS)*_n_* (*n* = 1–12) clusters is much weaker than Hg^0^ over the ZnS(110) surface. The reason may be that when Hg^0^ is adsorbed by (ZnS)*_n_* clusters, Hg^0^ only binds and interacts with the Zn atom in the (ZnS)*_n_* clusters; when Hg^0^ is adsorbed by ZnS(110), Hg^0^ binds and interacts with Zn and also binds with S on the ZnS(110) surface [8].

In order to study the reasons for the low adsorption energy of Hg^0^ on (ZnS)*_n_* (*n* = 1–12) clusters, we analyzed the Zn–Hg bond lengths in (ZnS)*_n_* (*n* = 1–12) clusters. Based on the geometry discussion section, the Zn–Hg bond lengths in (ZnS)*_n_*Hg range from 2.84–4.03 Å for (ZnS)*_n_*Hg (*n* = 3–12), while those for (ZnS)*_n_*Hg (*n* = 1–2) are 2.63 and 2.68 Å, respectively. The sum of the radius of Hg and Zn atoms is 2.94 Å according to the van der Waals radii by Bondi [34]. The Zn–Hg bond lengths in (ZnS)*_n_*Hg (*n* = 3–12) are near or bigger than the sum of the radius of Hg and Zn atoms, which indicates that interactions between Zn and Hg in the forming Zn–Hg bond are probably weak. The big steric hindrance in (ZnS)*_n_* may result in big bond lengths of Zn–Hg bonds in (ZnS)*_n_*Hg, particularly for *n* = 7–12. For *n* = 5 and 6, the very big Zn−Hg bonds are due to their particular structures of (ZnS)_5_Hg and (ZnS)_6_Hg. However, the bond lengths of Zn–Hg in (ZnS)*_n_*Hg (*n* = 1–2) are smaller than the sum of the radius of Hg and Zn atoms (2.94 Å) probably due to its small sizes of ZnS and (ZnS)_2_.

It can be seen from Figure 3b,c that when (ZnS)*_n_* clusters adsorb HgCl and HgCl_2_, the changing trends of *E*_ad_ with *n* are similar. For HgCl and HgCl_2_, when *n* increases from 1 to 12, *E*_ad_ tends to decrease. Overall, *E*_ad_ of HgCl adsorbed on (ZnS)*_n_* cluster is smaller than that of HgCl_2_. The adsorption energies of HgCl on (ZnS)*_n_* clusters range from −137.02 to −34.31 kcal/mol, while those of HgCl_2_ range from −233.11 to −117.27 kcal/mol with an exothermic process. Thus, the adsorption strength of HgCl_2_ over (ZnS)*_n_* clusters is much stronger than that of HgCl over (ZnS)*_n_* clusters. When *n* = 5 and 7, the adsorption energies are larger than that of the adjacent size. *E*_ad_ for (ZnS)_5_HgCl cluster is −106.54 kcal/mol, while (ZnS)_5_HgCl_2_ is −179.79 kcal/mol. *E*_ad_ values of HgCl and HgCl_2_ on (ZnS)_7_ cluster are −81.00 and −140.11 kcal/mol. When (ZnS)*_n_* (*n* = 1–12) clusters adsorb HgCl, HgCl decomposes into Hg and Cl atoms. This result can be interpreted as Hg and Cl atoms forming chemical bonds with S and Zn atoms in (ZnS)*_n_* clusters, respectively. When (ZnS)*_n_* (*n* = 1–12) clusters adsorb HgCl_2_, HgCl_2_ decomposes to HgCl and Cl. Then, the Hg atom in the HgCl part forms an Hg−S bond with the neighboring S atom in the (ZnS)*_n_* cluster, and the decomposed Cl atom forms a Zn−Cl bond with one Zn atom in the (ZnS)*_n_* cluster at a nearby position. The condition of HgCl and HgCl_2_ adsorption, here, is similar to those on the Fe_3_O_4_(111) surface [32]. Based on the adsorption energies and newly formed covalent Hg−S and Zn−Cl bonds, HgCl and HgCl_2_ adsorbed on (ZnS)*_n_* clusters are chemisorption. HgCl and HgCl_2_ are also chemisorption on the ZnS(110) surface with adsorption energies being −174.33 kJ/mol and −132.79 kJ/mol, respectively [8].

### 2.3. Non−Covalent Interactions (NCI) and Electron Local Function (ELF) Analyses

An ELF analysis supplies quantitative criteria to investigate the Jellium−like behavior in clusters and the changes in bonding properties at different regions. The ELF values are in the range from 0.0 to 1.0, where the highest value 1.0 (in red) indicates strong binding interaction, while the lowest value 0.0 (in blue) indicates weak interaction. We calculate the ELF values of (ZnS)*_n_*Hg (*n* = 5, 7) clusters and plot them in Figure 4. As shown in Figure 4, there is no bonding interactions between Hg and Zn. Therefore, what maintains the interaction between Zn and Hg? The NCI analysis of (ZnS)_5_Hg and (ZnS)_7_Hg is performed. The NCI analysis for (ZnS)_5_Hg and (ZnS)_7_Hg is also depicted in Figure 4. The intramolecular interactions can be distinguished from the figure. Non−bond attraction presents on the left side of the graph, while non−bond repulsion presents on the right side of the NCI graph. In Figure 4a, one attractive spike presents at −0.009 a.u. associated with the Zn···Hg interaction and one repulsive spike presents at 0.006 a.u. related to the Zn···Zn repulsive interactions in (ZnS)_5_Hg. The NCI isosurface of (ZnS)_5_Hg reveals that non−bond attractive interactions exist between Hg^0^ and its five neighboring Zn atoms in (ZnS)_5_. The non−bond attractive interactions play an important role in stabilizing the (ZnS)_5_Hg cluster. The ELF analysis of Hg^0^ and two neighboring Zn atoms is shown in Figure 4c.

From the figure, it is obvious that no Zn−Hg bond forms in (ZnS)_5_Hg. The Hg^0^ and Zn interaction with each other mainly comes from non−bond interactions. From Figure 4b, there is one attractive spike at −0.033 a.u. and two repulsive spikes at −0.006 and 0.030 a.u. that correspond to Zn···Hg and Zn···Zn repulsive interactions, respectively. The isosurface of the (ZnS)_7_Hg cluster in the middle of Figure 4 reveals that the attractive interactions are in blue, whereas the repulsive interactions are in red. It is worth noting that the Zn···Hg interaction in the (ZnS)_7_Hg cluster is stronger than the hydrogen bond in water dimer (−0.025 a.u.) [35]. Thus, the Zn···Hg interaction in the (ZnS)_7_Hg cluster is strong and can stabilize it. The three atoms’ ELF analysis of Hg and neighboring Zn and S atoms in Figure 4c reveals that the Zn−Hg bond does not form without red regions, while the Zn−S bond forms with red regions existing.

In order to determine why the adsorption energy of HgCl and HgCl_2_ on (ZnS)*_n_* clusters is relatively large, we carried out ELF and NCI analyses for (ZnS)_5_HgCl, (ZnS)_7_HgCl, (ZnS)_5_HgCl_2_, and (ZnS)_7_HgCl_2_ clusters with bigger adsorption energies. Figure 5 plots the NCI and ELF analyses for (ZnS)_5_HgCl and (ZnS)_7_HgCl clusters. The results, in Figure 5a, for (ZnS)_5_HgCl reveal two attractive spikes at −0.011 and −0.043 a.u. which are associated with the Zn···Zn and S···Hg attractive interactions, respectively. Two repulsive spikes, in Figure 5a, at 0.011 and 0.024 a.u. are Zn···Zn repulsive interactions in Zn_2_S_2_ and ZnSClZn units, respectively. The ELF analysis of Hg and neighboring S and Zn atoms is shown in the upper part of Figure 5c. According to the figure, it is obvious that the S−Hg bond forms in (ZnS)_5_HgCl because of the red region between Hg and S and the Zn−S bond also presents. The ELF analysis of Cl and two neighboring Zn atoms reveals that the Zn−Cl bond forms in (ZnS)_5_HgCl. Figure 5b shows the NCI analysis of the (ZnS)_7_HgCl cluster. From the figure, one attractive and two repulsive spikes present that are locating at −0.025, 0.007, and 0.029 a.u., which correspond to Hg···S attractive interactions and Zn···Zn repulsive interactions, respectively. The ELF analysis reveals that the Hg atom in the HgCl molecule forms an Hg−S bond with the S atom, and the Cl atom forms a Zn−Cl bond with the Zn atom in the (ZnS)_7_ cluster.

Figure 6 plots the NCI and ELF analyses for the (ZnS)_5_HgCl_2_ and (ZnS)_7_HgCl_2_ clusters. The NCI analysis for (ZnS)_5_HgCl_2_ in Figure 6a reveals two attractive spikes at −0.008 and −0.042 a.u. associated with the Zn···Zn and S···Hg attractive interactions, respectively. Two repulsive spikes, in Figure 6a, are located at 0.009 and 0.024 a.u. which are Zn···Zn repulsive interactions in Zn_2_S_2_ and ZnSClZn units, respectively. The ELF analysis of Cl and neighboring Hg and Zn atoms is shown in the upper part of Figure 6c. According to the figure, it is obvious that the Cl−Hg bond forms in (ZnS)_5_HgCl_2_ according to the red region between Hg and Cl. The ELF analysis of Hg and two neighboring S and Zn atoms reveals that Hg−S and Zn−S bonds form. The ELF analysis of Cl and two neighboring Zn atoms reveals that two Zn−Cl bonds form in the ZnSZnCl unit. Figure 6b is the NCI analysis of the (ZnS)_7_HgCl_2_ cluster. According to the figure, two attractive spikes are located at −0.022 and 0.005 a.u. which are the Zn···Zn and Hg···S attractive interactions, respectively; two repulsive spikes present at 0.005 and 0.029 a.u. which correspond to Zn···Zn repulsive interactions. The ELF analysis of Cl, Hg, and S reveals that the Hg atom forms Hg−Cl and Hg−S bonds in the (ZnS)_7_HgCl_2_ cluster. According to the ELF analysis of the remaining Cl and neighboring Zn and S atoms, the Cl−Zn bond forms. The ELF analysis of Hg and neighboring S and Zn reveals that the Hg−S bond forms in the (ZnS)_7_HgCl_2_ cluster.

In summary, Hg and Cl atoms in HgCl form Hg−S and Zn−Cl bonds with S and Zn atoms in (ZnS)_5_ and (ZnS)_7_ clusters, respectively. Meanwhile, Hg interacts with the neighboring S atom with non−bond interaction. For HgCl_2_ adsorption on (ZnS)_5_ and (ZnS)_7_ clusters, one Cl atom in HgCl_2_ forms a Zn−Cl bond with one Zn atom in (ZnS)_5_ and (ZnS)_7_ clusters. The Hg−Cl part of the HgCl_2_ molecule forms an Hg−S bond with one S atom in (ZnS)_5_ and (ZnS)_7_ clusters. The newly formed Hg−S and Zn−Cl bonds and non−bond interactions stabilize (ZnS)_5_HgCl, (ZnS)_7_HgCl, (ZnS)_5_HgCl_2_, and (ZnS)_7_HgCl_2_, and thus, the adsorption energies of them are large. When Hg^0^ adsorbed on (ZnS)_5_ and (ZnS)_7_ clusters, only non−bond interactions present between Hg^0^ and S atoms, no obvious Hg−S bond forms. Thus, the adsorption energies for Hg^0^ on (ZnS)n clusters are much smaller than those of HgCl and HgCl_2_.

### 2.4. Projected Density of State (PDOS) Analysis

In order to obtain the interaction behaviors between the (ZnS)*_n_* cluster and Hg, HgCl, and HgCl_2_ molecules, we take the projected density of states (PDOS) of (ZnS)_7_Hg, (ZnS)_7_HgCl, and (ZnS)_7_HgCl_2_ as examples. Figure 7 shows the PDOS maps of bare Hg^0^ when the adsorption does not happen, and the Hg and Zn atoms of the Zn−Hg bond in (ZnS)_7_Hg.

From Figure 7a, the characteristic energies of *s*−orbitals of isolated Hg are at −7.56 and 4.57 eV, and *p*−orbitals are characterized at −0.43 and 4.28 eV. The energy of *d*−orbitals of Hg locates at −11.71 eV. In Figure 7b of the (ZnS)_7_Hg cluster, the energy of Hg−*s* orbitals shifts to −10.17/−8.59 eV, and the energy of Hg−*p* orbitals locate at −1.13/0.86 eV. The energy of *d*−orbitals is at −13.04 eV. By comparing the characteristic values of Hg atomic orbital energy before and after adsorption, it can be seen that the atomic orbitals of Hg^0^ move to lower energy positions. From Figure 7b, we can observe, obviously, the orbital hybridization phenomenon. Zn−*d* and Hg−*d* orbital hybridizations occur at −13.39 eV; Zn−*d*, Hg−*d*, and Zn−*p* hybridizations occur at −9.63 eV; Zn−*p* and Hg−*p* hybridizations occur at 1.04 eV. The hybridization confirms electrons transfer between Zn and Hg atoms in (ZnS)_7_Hg. The Mulliken electron population analysis reveals that the charges of Hg and Zn atoms are 0.140 and 0.540 |e|, respectively. Figure 8 presents the PDOS maps of: atoms in bare HgCl, the Hg and S atoms of the Hg−S bond in (ZnS)_7_HgCl, and the Cl and Zn atoms of the Cl–Zn bond in (ZnS)_7_HgCl. It is worth noting that both bare HgCl and (ZnS)_7_HgCl are doublet molecules and each of them has one unpaired electron. Thus, the PDOS maps of bare HgCl and (ZnS)_7_HgCl include both α and β style molecular orbitals. From Figure 8a of bare HgCl, *s*−orbitals of Hg locate at −11.18/−2.29 eV, *p*−orbitals at −0.65/3.06 eV, and *d*−orbitals at −14.64 eV. When free HgCl is absorbed on the (ZnS)_7_ cluster, Hg−S and Zn−Cl bonds form in the (ZnS)_7_HgCl compound. According to the Mulliken electron population analysis, the charges of Hg and S are 0.266 and −0.481 |e|, respectively, and the charges of Zn and Cl are 0.490 and −0.382 |e|, respectively. Thus, charge transfer occurs from Hg to S for the Hg−S bond and from Zn to Cl for the Zn−Cl bond. From Figure 8a, α *s*−orbitals of Hg is at −10.52/−7.25 eV before adsorption. After adsorption, the peaks of α *s*−orbitals of Hg in (ZnS)_7_HgCl change little, which are at −10.46 and −6.40 eV, respectively. By comparing the peaks of α *p*−orbitals before and after adsorption, it can be observed that the peak has a small change, from −13.44 to −13.19 eV. The peaks of α *d*−orbitals also change little, which change from −0.90 and 3.46 to −0.93 and 3.23 eV, respectively. From Figure 8b, hybridization phenomena are present at −10.14, −7.01, −5.80 eV, and all of them are α S−*p* and α Hg−*s* orbital hybridizations. The hybridizations of α Zn−*s* and α Cl−*p* orbitals occurs at −11.29 eV; α Zn−*s* and Zn−*p* orbitals hybrids at −1.13 eV. According to Figure 8a,c, the peaks of β Cl−*p* are at −8.42 and −9.73 eV before adsorption, whereas they are at −8.49 and −10.70 eV after adsorption. The hybridizations of β Cl−*p* and β Zn−*s* orbitals occur at −11.39 eV; the hybridization of β Zn−*s* and Zn−*p* orbitals occur at −1.29 eV. From Figure 8a,c, the peaks of β Cl−*p* are at −8.42 and −9.73 eV before adsorption, whereas they are at −8.49 and −10.70 eV after adsorption. The hybridizations of β Cl−*p* and β Zn−*s* orbitals occur at −11.39 eV; the hybridization of β Zn−*s* and Zn−*p* orbitals occur at −1.29 eV. Figure 9 includes three PDOS maps: the PDOS map of Hg and one Cl atom in the HgCl_2_ molecule; the PDOS map of Hg and S atoms of the Hg−S bond in (ZnS)_7_HgCl_2_; the PDOS map of Cl and Zn atoms of the Cl−Zn bond in (ZnS)_7_HgCl_2_. From Figure 9a, the characteristic peaks of *s*−orbitals for Hg are at −11.18/−2.29 eV, the characteristic peaks of *p*−orbitals are at −0.65/3.06 eV, and the characteristic peak of *d*−orbitals is at −14.64 eV in the bare HgCl_2_ molecule. When (ZnS)_7_HgCl_2_ forms, Hg−S and Zn−Cl bonds form. From Figure 9b, the charges of Hg and S are 0.634 and −0.483 |e|, and those of Zn and Cl atoms are 0.558 and −0.394 |e|, indicating that charges transferred from Hg to S and from Zn to Cl for Hg−S and Zn−Cl bonds, respectively. The Hg−*s* orbitals in (ZnS)_7_HgCl_2_ are at −11.16/−2.04 eV, the Hg−*p* orbitals are at −0.61/−1.91 eV, and the Hg−*d* orbitals are at −14.22. As compared with the characteristic energies of the orbitals of Hg in free HgCl_2_, the orbital energy levels of Hg in (ZnS)_7_HgCl_2_ changes little. From Figure 9b, Hg−*s* and S−*p* orbital hybridizations exists at −11.22 eV, and Hg−*s*, Hg−*p* and S−*p* orbital hybridizations present at −1.51 eV in (ZnS)_7_HgCl_2_. The Cl−*p* orbitals in free HgCl_2_ molecule (Figure 9a) are at −15.21/−9.32/−2.28 eV, while in (ZnS)_7_HgCl_2_ they are at −14.49/−8.58/−2.03 eV (Figure 9c). According to Figure 9c, orbital hybridizations of Cl−*p* and Zn−*p* orbitals occur at −14.95/−9.35 eV, while Zn−*s* and Cl−*p* orbital hybridizations are at −10.61/1.65 eV for (ZnS)_7_HgCl_2_. The orbital hybridizations of Zn and Cl prove the bond interaction between them in the (ZnS)_7_HgCl_2_ molecule.

### 2.5. Stability

Figure 10 presents the relative stability analysis. Figure 10a shows the second−order energy differences (Δ_2_*E*) versus cluster size and Figure 10b shows the plots of the energetic gaps (*E*_at_–*E*_ave_) of (ZnS)*_n_*Hg, (ZnS)*_n_*HgCl, and (ZnS)*_n_*HgCl_2_ (*n* = 1–12) clusters as a function of cluster size *n*, where *E*_at_ is the atomization energy and *E*_ave_ is average energy. For (ZnS)*_n_*Hg, *E*_ave_ = −4163.15552 − 22.15309 × *n*^1/3^ + 15.08148 × *n*^2/3^ − 59251.4955 × *n*. For (ZnS)*_n_*HgCl, *E*_ave_ = −16705.92608 + 16.58575 × *n*^1/3^ − 6.2664 × *n*^2/3^ − 59247.58793 × *n*. For (ZnS)*_n_*HgCl_2_, *E*_ave_ = −29236.31971 + 36.76056 × *n*^1/3^ − 19.12771 × *n*^2/3^ − 59244.98538 × *n*. The rules of their stability changing as *n* is given in Figure 10a. In cluster science, Δ_2_*E* is a parameter reflecting cluster stability. The higher the value of Δ_2_*E*, the more stable the cluster is. There are four peaks in Figure 10a corresponding to *n* = 2, 5, 7, and 10, manifesting that the clusters at these sizes are more stable than the neighboring sizes for (ZnS)*_n_*Hg, (ZnS)*_n_*HgCl, and (ZnS)*_n_*HgCl_2_. In addition, there are four downward peaks at *n* = 3, 6, 9, and 11, which indicate that they are less stable sizes for (ZnS)*_n_*Hg, (ZnS)*_n_*HgCl, and (ZnS)*_n_*HgCl_2_.

In Figure 10b, the size at the upward peak is more stable, and that at the downward peak is less stable. There are upward peaks at *n* = 2, 5, 7, and 10, representing that they are more stable than their neighbor sizes. There are downward peaks at *n* = 3, 6, and 9, which indicates that they are less stable.

Not only from the results of Δ_2_*E*, but also from those of *E*_at_ − *E*_ave_, (ZnS)*_n_*Hg, (ZnS)*_n_*HgCl, and (ZnS)*_n_*HgCl_2_ clusters are relative more stable sizes at *n* = 2, 5, 7, and 10.

## 3. Materials and Methods

### Computational Methods

The structures of (ZnS)*_n_* clusters were investigated by using the genetic algorithm combined with density functional theory (GA−DFT) method. The GA algorithm is a search heuristic algorithm that simulates the evolution process [36]. A concrete description of the GA−DFT method has been given in our former paper [37]. In the GA−DFT program, the initial search of clusters is conducted at the PBE0/def2−SVP level [38,39]. After checking the similarity of the searched structures, the top 20 low−energy isomers in the structure library are optimized at the PBE0/def2−TZVP level. The isomers are arranged according to the energy order. The structure with the lowest energy is the global minimum (GM) structure of the cluster. The PBE0/def2−TZVP method [38,39] has been widely used in zinc compounds [40,41,42,43,44]. The structures of (ZnS)*_n_*Hg, (ZnS)*_n_*HgCl, and (ZnS)*_n_*HgCl_2_ clusters have been obtained based on the GM structures of (ZnS)*_n_*. Concretely, isomers with different structures are obtained by adding an Hg atom, HgCl, and HgCl_2_ to the GM structures of (ZnS)*_n_* clusters, and the isomers with the lowest energies are the global optimal structures of (ZnS)*_n_*Hg, (ZnS)*_n_*HgCl, and (ZnS)*_n_*HgCl_2_ clusters, respectively. In the computational processes of (ZnS)*_n_*, (ZnS)*_n_*Hg, and (ZnS)*_n_*HgCl_2_, the charge and density are set to zero and one, respectively. The charge and density are set to zero and two for (ZnS)*_n_*HgCl clusters. For Zn, S, and Cl atoms, all the electrons are included in the calculations, whereas an effective core potential (ECP) and 20 valence electrons are considered for the Hg atom. Vibrational energies of each structure are computed to verify whether it is a stable structure. All the structures reported here are stable without one imaginary frequency. The D3(BJ) correction of van der Waals force is used in geometric optimization and vibration frequency calculation processes [45]. The D3 balance correction method proposed by Boys and Bernadi can correct the calculated lower binding energy value, thus, effectively making up for the precision defect of the PBE0 method in the research process [34]. All calculations are performed in the Gaussian 16 program [46]. The convergence threshold parameters of Gaussian 16 during optimization are 4.5 × 10^−4^ for maximum force, 3 × 10^−4^ for RMS force, 1.8 × 10^−4^ for maximum displacement, and 1.2 × 10^−4^ for RMS displacement. When geometric optimization is performed, the atoms of Hg^0^, HgCl, and HgCl_2_ are completely relaxed. The Gauss view 6.0.16 visualization software is used to obtain the molecular structures [47], and all the structures of the clusters are viewed and obtained through the software. The Multiwfn 3.8 software [48] is used to calculate electron localization function (ELF), non−covalent interactions (NCI), and projected density of state.

According to the following formula [33], the adsorption energy (*E*_ads_) is obtained:(1)$$E_{\mathrm{ad}}=E_{\left(adsorbate/substrate\right)}-(E_{adsorbate}+E_{substrate})$$
where *E*_(*adsorbate*/*substrate*)_ represents the energy of (ZnS)*_n_*Hg, (ZnS)*_n_*HgCl, or (ZnS)*_n_*HgCl_2_; *E_adsorbate_* is the energy of Hg^0^, HgCl, or HgCl_2_; *E_substrate_* is the energy of (ZnS)*_n_* clusters.

All the energies are those for the global minimum structures. Basis set superposition error (BSSE) has been considered during the adsorption energy (*E*_ad_) calculations of Hg^0^, HgCl, and HgCl_2_ over (ZnS)*_n_* clusters. If the value of *E*_ads_ is negative, the adsorption process is exothermic and it favors to occur. When studying adsorption, it is particularly important to distinguish between physical adsorption and chemical adsorption of adsorbate. Whether new chemical bonds are formed is an important basis for distinguishing physical adsorption from chemical adsorption. In this study, chemisorption is judged by two points: new chemical bonds form between the adsorbate and the substrate during adsorption; the adsorption energy is relatively large.

The non−covalent interaction (NCI) analysis method can be used to analyze the stability of clusters. The NCI method [35] proposed by Yang et al. has been used to analyze many systems [49,50,51]. During the NCI analysis, the function is reduced density gradient (RDG) and the variable is electron density (ρ). The expression of RDG is:(2)$$s=\frac{1}{2{\left(3\pi^{2}\right)}^{1/3}}\frac{\left|\nabla \rho \right|}{\rho^{4/3}}$$

The electron density Hessian matrix (i.e., sign(*λ*_2_)) can be used to differentiate bonding (λ_2_ < 0) and non−bonded (λ_2_ > 0) interactions. The NCI diagram is a scatter plot, where the ordinate is *s* (RGB) and the abscissa is *ρ* * sign (*λ*_2_)_._ Low−density peaks in the NCI map stand for non−covalent interactions. Using the VMD program, the gradient isosurface can be obtained [52]. Based on the values of sign(λ_2_) * ρ, the gradient isosurfaces are colored [52] according to a RGB (red−blue−green) scale.

To obtain the relative stability of (ZnS)*_n_*Hg, (ZnS)*_n_*HgCl, and (ZnS)*_n_*HgCl_2_ clusters with different sizes, second−order difference energy (Δ_2_*E*) values of the GM structures are calculated. The formula of Δ_2_*E* is as follows:Δ_2_*E* = *E*_[(ZnS)*n*−1A]_ + *E*_[(ZnS)*n+*1A]_ − 2*E*_[(ZnS)*n*A]_(3)

In the formula, A represents Hg, HgCl, and HgCl_2_. *E*_[(ZnS)*n*−1A]_, *E*_[(ZnS)*n+*1A]_, and *E*_[(ZnS)*n*A]_ are the energies of (ZnS)*_n−_*_1_A, (ZnS)*_n+_*_1_A, and (ZnS)*_n_*A, respectively.

Average energies (*E*_ave_) of the GM structures for the (ZnS)*_n_*Hg, (ZnS)*_n_*HgCl, and (ZnS)*_n_*HgCl_2_ series are also calculated to study the relative stability of the clusters. *E*_ave_ are four−parameter fitting functions of the GM energies, which can be expressed as: *E*_ave_ = a + b × *n*^1/3^ + c × *n*^2/3^ + d × *n*, where *n* is the size of the cluster; a, b, and c are fitting coefficients. *E*_at_ is the GM energies of the clusters.

## 4. Conclusions

Hg^0^ is able to be physisorbed on (ZnS)*_n_* (*n* = 1–12) clusters with adsorption energy in the range from −13.33 to −4.04 kcal/mol. The orbitals of Hg^0^ can hybridize with the orbitals of Zn. The interactions between (ZnS)*_n_* (*n* = 1–12) and Hg^0^ are mainly non−convalent interactions. HgCl and HgCl_2_ can interact with the S and Zn to form Hg−S and Zn−Cl bonds, which induce strong chemisorptions on (ZnS)*_n_* (*n* = 1–12) clusters. Considering that the adsorption energies of HgCl and HgCl_2_ on (ZnS)*_n_* (*n* = 1–12) clusters predicted in this study are large and new chemical bonds are formed between HgCl, HgCl_2_, and (ZnS)*_n_* clusters, we believe that (ZnS)*_n_* clusters have great potential as efficient mercuric chloride capture materials for coal−fired power plants. Our future work will focus on the reaction mechanism of Hg^0^, HgCl, and HgCl_2_ adsorption on (ZnS)*_n_* clusters, which is another important issue of Hg^0^, HgCl, and HgCl_2_ capture. We will also consider various zinc−related materials, including pure zinc, to further explore the role of sulfur in enhancing Hg^0^, HgCl, and HgCl_2_ binding.

## Figures and Tables

**Figure 1 molecules-28-01214-f001:**
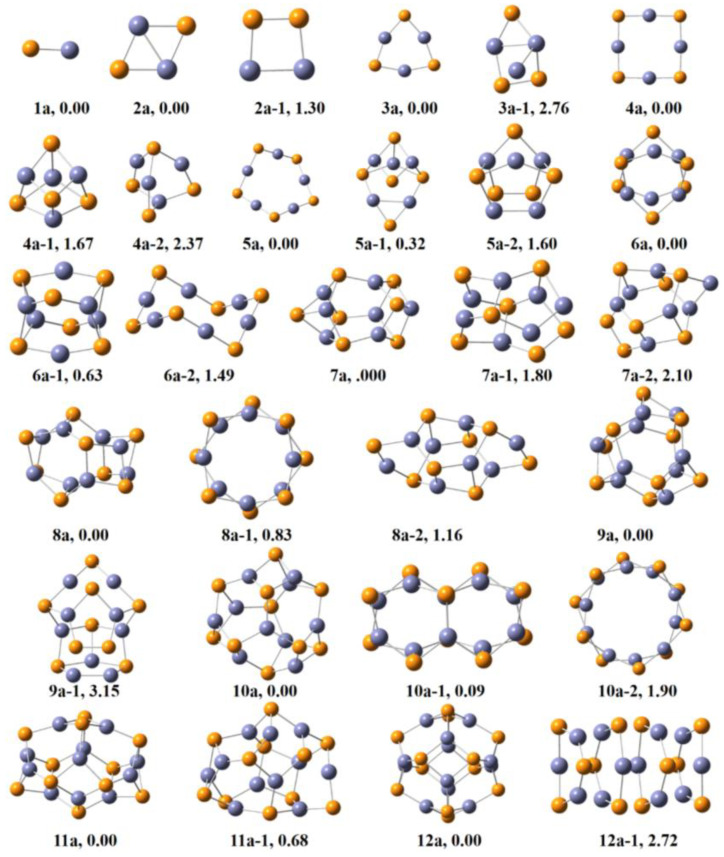
The global minimum structures (GMs) and low−lying isomers of (ZnS)*_n_* clusters at the PBE0−D3BJ/def2−TZVP level of theory. GM structures are na; na−1 and na−2 are low−lying isomers (lavender, Zn and S, yellow).

**Figure 2 molecules-28-01214-f002:**
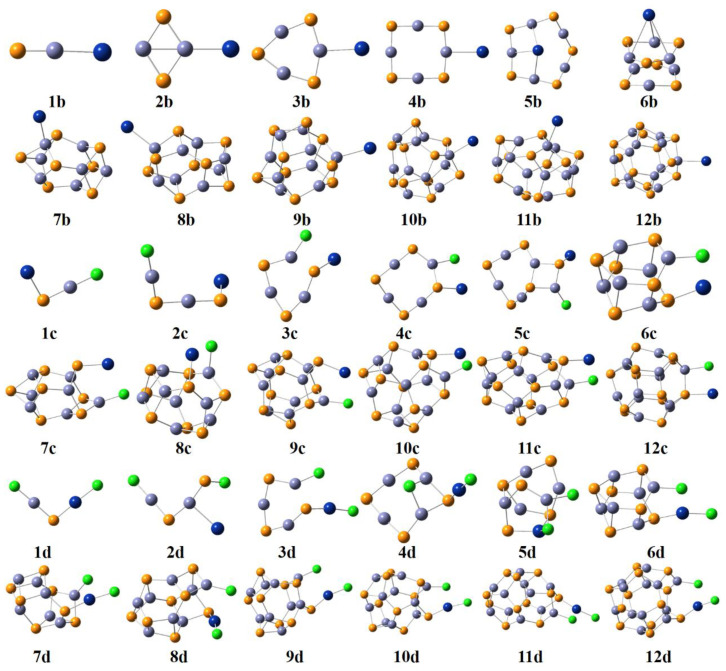
The global minimum structures (GMs) at the PBE0−D3BJ/def2−TZVP level of theory. *n*b, (ZnS)*_n_*Hg; *n*c, (ZnS)*_n_*HgCl; *n*d, (ZnS)*_n_*HgCl_2_. (The lavender, yellow, dark blue, and green spheres represent Zn, S, Hg, and Cl atoms, respectively.)

**Figure 3 molecules-28-01214-f003:**
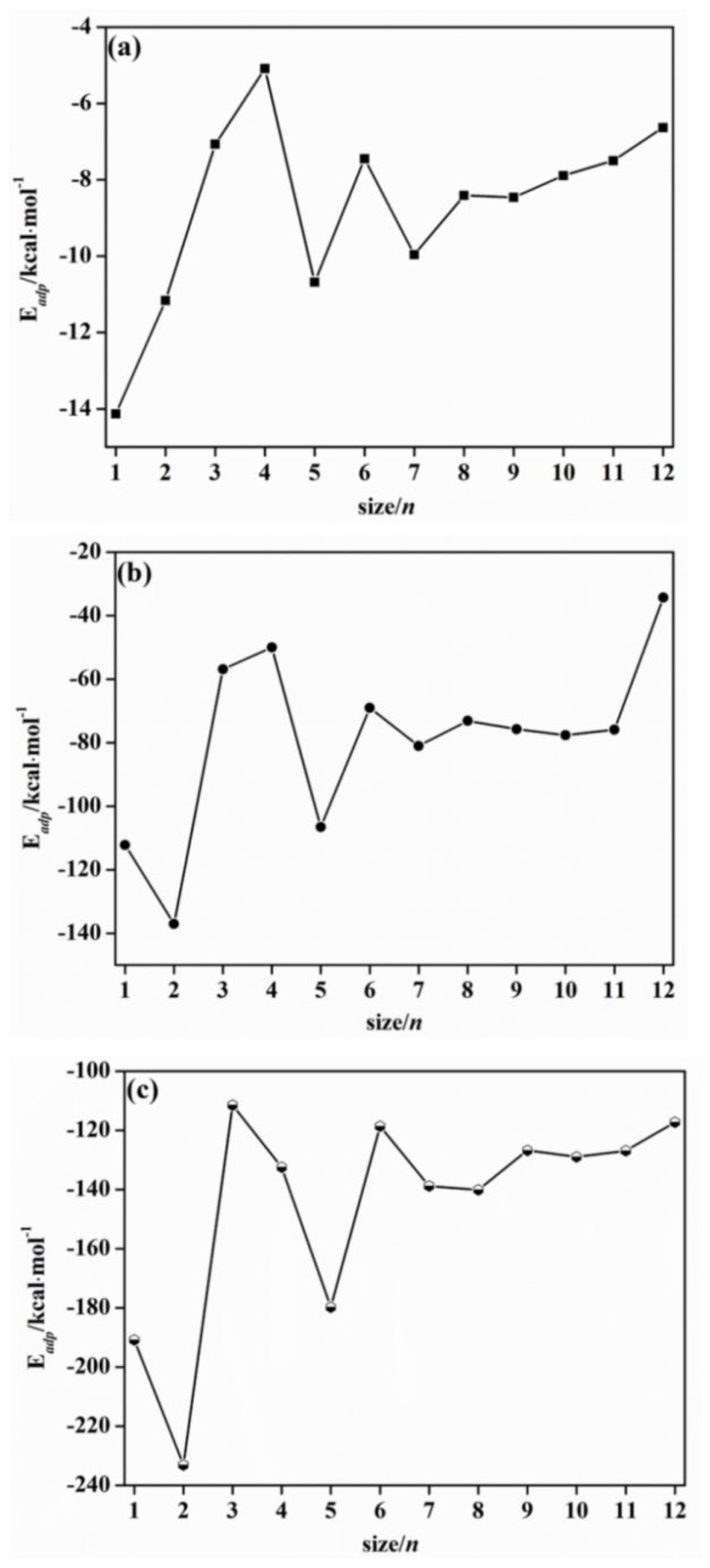
The adsorption energies of: (**a**) (ZnS)*_n_*Hg^0^; (**b**) (ZnS)*_n_*HgCl; (**c**) (ZnS)*_n_*HgCl_2_ (*n* = 1−12) clusters versus cluster size (*n*).

**Figure 4 molecules-28-01214-f004:**
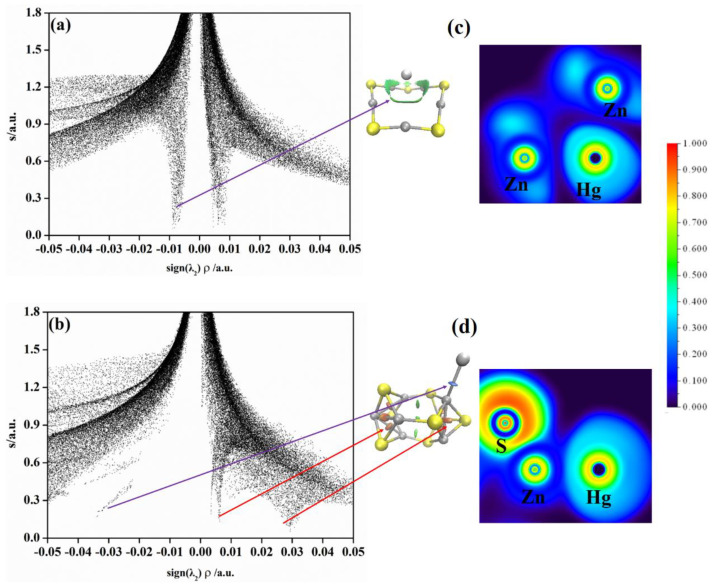
(**a**) Plots of s versus sign(λ2)ρ (left) and NCI isosurfaces (s = 0.25) (right) for (ZnS)_5_Hg; (**b**) plots of s versus sign(λ2)ρ (left) and NCI isosurfaces (s = 0.25) (right) for (ZnS)_7_Hg (Zn, gray; S, yellow; Hg, silvery); (**c**) the ELF contours for Zn(8)–Hg(11)–Zn(9) and (**d**) Hg(15)–Zn(13)–S(8) for (ZnS)_5_Hg and (ZnS)_7_Hg clusters. Labeled is the color scale of the values.

**Figure 5 molecules-28-01214-f005:**
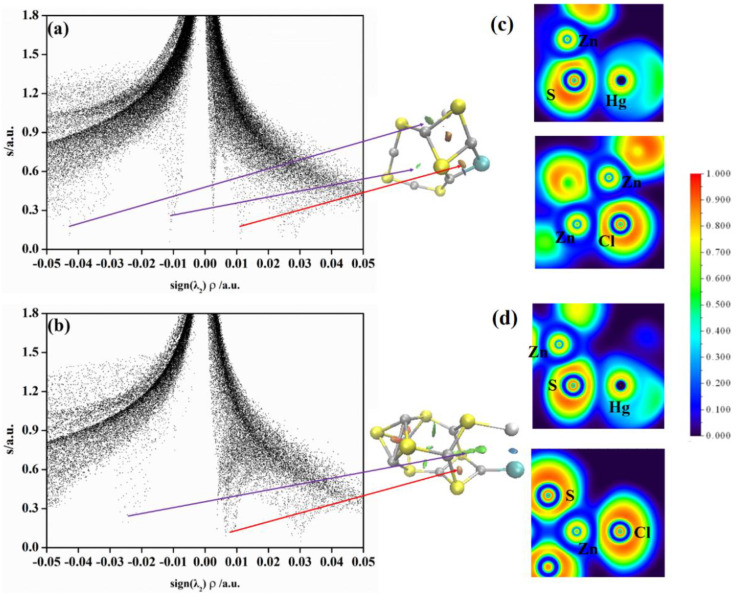
(**a**) Plots of s versus sign(λ2)ρ (left) and NCI isosurfaces (s = 0.25) (right) for (ZnS)_5_HgCl; (**b**) plots of s versus sign(λ2)ρ (left) and NCI isosurfaces (s = 0.25) (right) for (ZnS)_7_HgCl (Zn, gray; S, yellow; Hg, silvery; Cl, azure); (**c**) the ELF contours for Hg(12)–S(6)–Zn(7) and Cl(11)–Zn(7)–Zn(2) for (ZnS)_5_HgCl; (**d**) the ELF c ontours for Hg(15)–S(9)–Zn(2) and Cl(16)–Zn(6)–S(5) for (ZnS)_7_HgCl clusters. Labeled is the color scale of the values.

**Figure 6 molecules-28-01214-f006:**
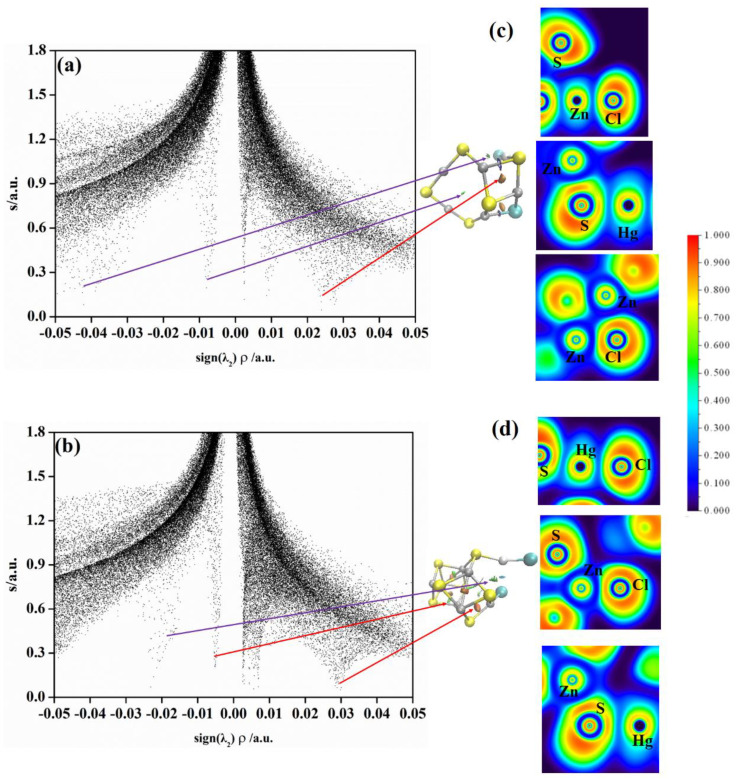
(**a**) Plots of s versus sign(λ2)ρ (left) and NCI isosurfaces (s = 0.25) (right) for (ZnS)_5_HgCl_2_; (**b**) plots of s versus sign(λ2)ρ (left) and NCI isosurfaces (s = 0.25) (right) for (ZnS)_7_HgCl_2_ (Zn, gray; S, yellow; Hg, silvery; Cl, azure); (**c**) the ELF contours for Cl(13)–Zn(12)–S(4), Hg(12)–S(6)–Zn(7), and Cl(11)–Zn(7)–Zn(2) for the (ZnS)_5_HgCl_2_ cluster; (**d**) the ELF contours for Cl(23)–Hg(22)–S(4), Cl(21)–Zn(5)–S(6), and Hg(22)–S(4)–Zn(7) for the (ZnS)_7_HgCl_2_ cluster. Labeled is the color scale of the values.

**Figure 7 molecules-28-01214-f007:**
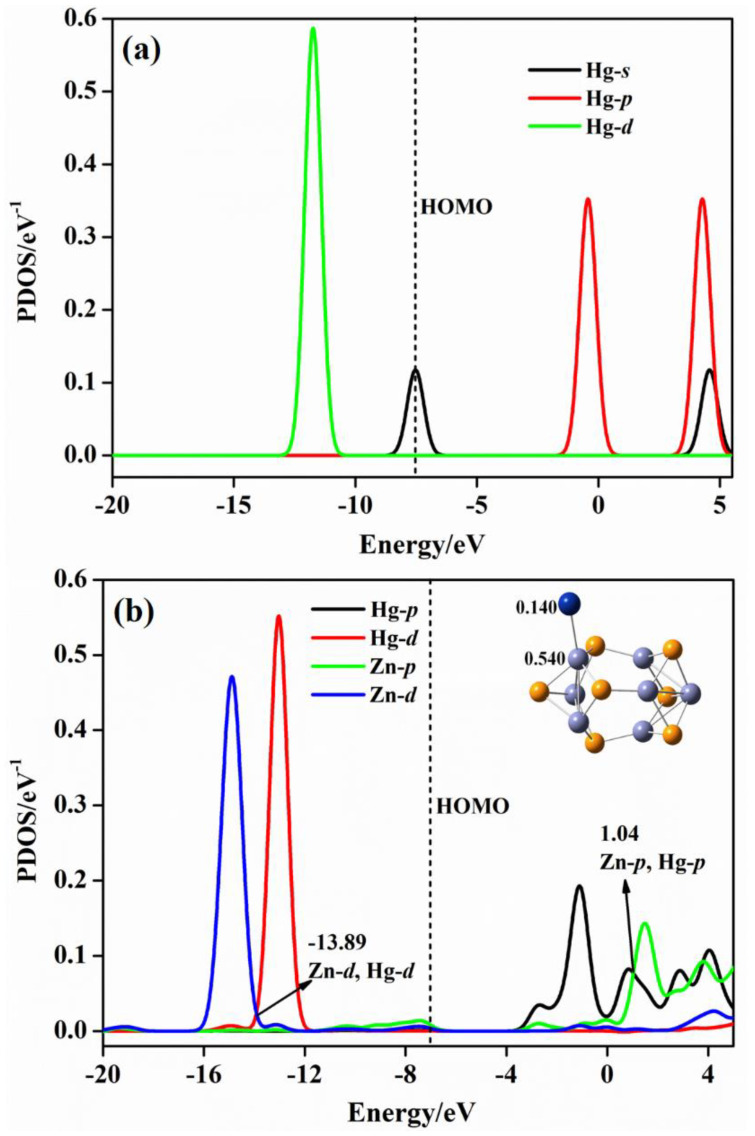
PDOS of: (**a**) Bare Hg^0^; (**b**) Hg and Zn atoms of the Zn–Hg bond in (ZnS)_7_Hg. The vertical dashed lines shows the positions of HOMOs.

**Figure 8 molecules-28-01214-f008:**
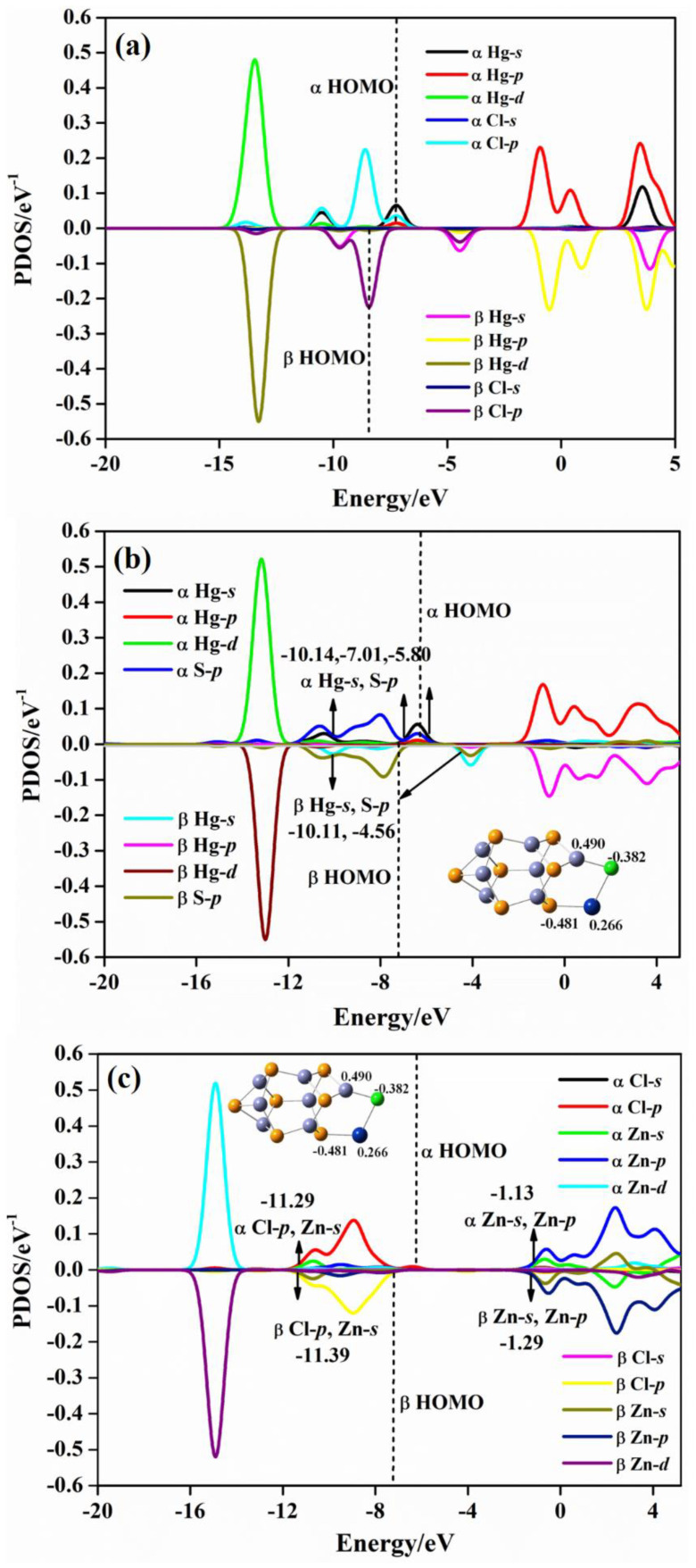
PDOS of: (**a**) Atoms in bare HgCl; (**b**) Hg and S atoms of the Hg–S bond in (ZnS)_7_HgCl; (**c**) Cl and Zn atoms of the Cl–Zn bond in (ZnS)_7_HgCl.

**Figure 9 molecules-28-01214-f009:**
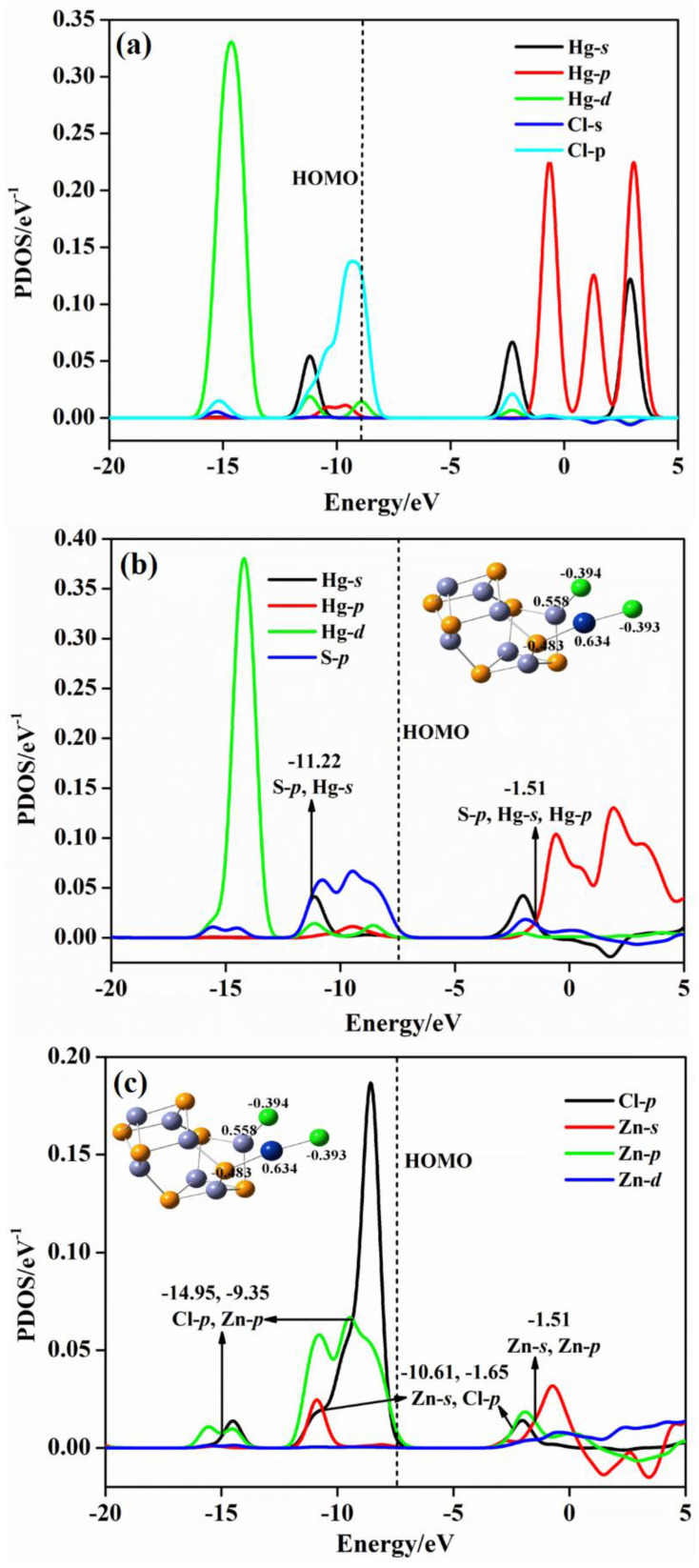
PDOS of: (**a**) Atoms in bare HgCl_2_; (**b**) Hg and S atoms of the Hg–S bond in (ZnS)_7_HgCl_2_; (**c**) Cl and Zn atoms of the Cl–Zn bond in (ZnS)_7_HgCl_2_.

**Figure 10 molecules-28-01214-f010:**
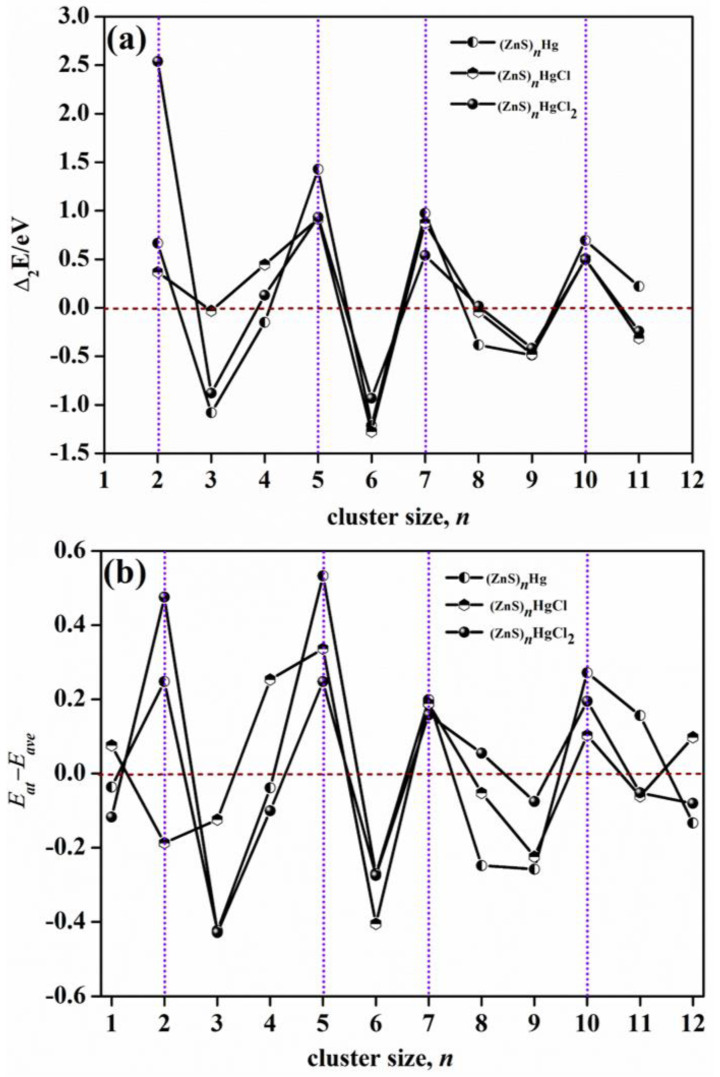
Relative stability analysis: (**a**) The second–order energy differences (Δ_2_*E*) versus cluster size, *n*; (**b**) plots of the energetic gaps (*E*_at_–*E*_ave_) of (ZnS)*_n_*Hg, (ZnS)*_n_*HgCl, and (ZnS)*_n_*HgCl_2_ (*n* = 1–12) clusters as a function of cluster size *n*, where *E*_at_ is the atomization energy and *E*_ave_ is the average energy. (ZnS)*_n_*Hg: *E*_ave_ = −4163.15552 − 22.15309 × *n*^1/3^ + 15.08148 × *n*^2/3^ − 59251.4955 × *n*; (ZnS)*_n_*HgCl: *E*_ave_ = −16705.92608 + 16.58575 × *n*^1/3^ − 6.2664 × *n*^2/3^ − 59247.58793 × *n*; (ZnS)*_n_*HgCl_2_: *E*_ave_ = −29236.31971 + 36.76056 × *n*^1/3^ − 19.12771 × *n*^2/3^ − 59244.98538 × *n*.

## Data Availability

Not applicable.

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
