# Peer review of "Density Functional Study to Investigate the Ability of (ZnS)n (n = 1–12) Clusters Removing Hg0, HgCl, and HgCl2 via Electron Localization Function and Non−Covalent Interactions Analyses"

_molecules, 2023, doi:10.3390/molecules28031214_

Round 1
Reviewer 1 Report
Tian Z. Et al. Studied the structures and bond properties between neutral (ZnS)n (n = 1–12) clusters and Hg0, HgCl, HgCl2by using PBE0-D3BJ/def2-TZVP density functional method. They analysed geometries of (ZnS)n (n = 1-12) clusters, they maid non-covalent interactions (NCI) and electron local function (ELF) , and Projected density of state (PDOS) , and stability analysis. The authors hope that the calculations can supply some information for (ZnS)n cluster adsorption. Thus, some points must be corrected and updated:
1. The introduction ends (line 79) with the sentence „The research results here are of great significance in the aspects of adsorption mechanism and adsorbents design.“ The authors must be list the aspects they think they are significant.
2. „Computational methods“ explain that calculation were perfomeed by using PBE0/def2-TZVP. I could not find the motivation of using this computational level. The authors should make small literature review about the used methods.
3. In line 94 there is need to add citations: at the moment it is writed that „reported in the litratures“ but there are no references.
4. Page 3. Line 98. The scentece with Gaussview should be removed. The authors are not citing other packages whch were used for writing the manuscript. I could not find the methods which were are inplemented in Gaussview either.
5. Line 100-110. I think this theory is from literature and it must be cited.
6. Question for „Computational methods“. How was the clusters formed? I could not find in other places of manucript the procdures either. It must be explained.
7. Line 135-136. It should be discussed at what levele the strcuture 6a-1 were predicted in ref. 30. Hwo it differ from the authors one.
8. Page 4. Line 151. The structure 12a and reference 35 is confusing. The scentence must be clearified: what is the result and how it relates to ref. 35?
9. Page 4. Line 155. References [23,37,38] and scentence is confusing as the previous scentences cite ref 35 and ref 36. It must be rewriten as now it is unlcear what the authors want to say.
10. Page 6. Line 199. It is unlcear how the ref 8 relates with the results.
11. Page 4, Line 235. The BSSE information must be moved to „Computational methods“ section.
12. Page 4. Line 239. It is unclear why it is chemical adsorbtion and why ref 41 is cited.
13 Page 4. Line 258. What does it mean Ref. 42: is it authors results or from references?
14. Page 4. Line 276. What does it mean Ref. 8: is it authors results or from references?
14. Page 9. Line 290-291. The first scentece is said already. Why do you repeat?
15. Page 11 Line 328. Shoudd it be „how“ instead of the „why“?
16. Page 14. Figure 7. The quality of figures is very low. It must be enlarged.
17. Page 15. Line 429-454. The computational description parts must be moved to „Computational methods“ section.
18. Page 16. Line 441-446. The figure discription must ne moved to manuscript text.
19. Page 17. Line 471. I‘m not sure why the authors expect other to find some information form others in their results. Instead could the authors list the important information by theireselft for the reader?
Reviewer 2 Report
Pls find attached file

Round 2
Reviewer 1 Report
It can be published in present form.
Reviewer 2 Report
The revised manuscript can be considered for publication.